# Conditional Ablation of Spred1 and Spred2 in the Eye Lens Negatively Impacts Its Development and Growth

**DOI:** 10.3390/cells13040290

**Published:** 2024-02-06

**Authors:** Fatima Wazin, Frank J. Lovicu

**Affiliations:** 1Molecular and Cellular Biomedicine, School of Medical Sciences, Faculty of Medicine and Health, The University of Sydney, Sydney, NSW 2006, Australia; fatima.wazin@gmail.com; 2Save Sight Institute, Faculty of Medicine and Health, The University of Sydney, Sydney, NSW 2006, Australia

**Keywords:** lens development, morphogenesis, Spred, Sprouty, RTK, ERK1/2, FGF, proliferation, fiber differentiation, microphakia, microphthalmia

## Abstract

The development and growth of the eye depends on normal lens morphogenesis and its growth. This growth, in turn, is dependent on coordinated proliferation of the lens epithelial cells and their subsequent differentiation into fiber cells. These cellular processes are tightly regulated to maintain the precise cellular structure and size of the lens, critical for its transparency and refractive properties. Growth factor-mediated MAPK signaling driven by ERK1/2 has been reported as essential for regulating cellular processes of the lens, with ERK1/2 signaling tightly regulated by endogenous antagonists, including members of the Sprouty and related Spred families. Our previous studies have demonstrated the importance of both these inhibitory molecules in lens and eye development. In this study, we build on these findings to highlight the importance of Spreds in regulating early lens morphogenesis by modulating ERK1/2-mediated lens epithelial cell proliferation and fiber differentiation. Conditional loss of both Spred1 and Spred2 in early lens morphogenesis results in elevated ERK1/2 phosphorylation, hyperproliferation of lens epithelia, and an associated increase in the rate of fiber differentiation. This results in transient microphakia and microphthalmia, which disappears, owing potentially to compensatory Sprouty expression. Our data support an important temporal role for Spreds in the early stages of lens morphogenesis and highlight how negative regulation of ERK1/2 signaling is critical for maintaining lens proliferation and fiber differentiation in situ throughout life.

## 1. Introduction

Spreds (Sprouty-related Ena/VASP homology-1-domain-containing proteins) are membrane-associated antagonists of growth factor-induced ERK/MAPK activation [1,2], implicated in regulating cellular behavior and development. Spred1, Spred2, and Spred3 are among the identified mammalian Spred homologues, characterized by their distinct structural features and roles in various developmental processes [2]. Their antagonistic effects are extensive, having been demonstrated to regulate various processes like hematopoiesis, eosinophil proliferation and activation, bone morphogenesis, lymphatic vessel development, and eye development [3,4,5,6,7,8]. Moreover, they have been implicated in the pathogenesis of various diseases including cancer, as well as pulmonary and cardiovascular disease [9,10,11,12,13,14]. Although there has been extensive research into other related MAPK antagonists, such as Sprouty, in various organs including the eye, there is limited knowledge of Spreds’ precise mechanism of action, particularly in regulating growth factor signaling pathways [1].

In lens development, growth factor-mediated signaling via the MAPK/ERK1/2 pathway (the most abundant MAPK in the lens) is important in regulating lens epithelial cell proliferation and fiber differentiation [10,15,16,17]. Earlier studies have shown that the onset, duration, and levels of ERK1/2 activity are important for determining lens cell fate and cellular activity, highlighting the need for its tight regulation [7,8,15,17,18]. The tight regulation of such signaling pathways is achieved through negative feedback processes, modulating growth factor stimulation to ensure a consistent physiological response [19]. Spred proteins, along with Sprouty and Sef, have been identified as negative regulators of receptor tyrosine kinase (RTK) signaling and have been demonstrated to influence cell fate decisions throughout lens development by modulating growth factor stimulation [2,7,8,20,21,22,23]. Given Spred’s unique membrane-associated inhibition of ERK1/2-signaling and its co-expression with other established RTK antagonists such as Sprouty (and even Sef) in the lens, we were interested in determining its role in lens development and growth [22,24], including its potential interplay and relationship with the other regulators of RTK signaling. 

Recent studies in our laboratory have reported Spreds in regulating lens cell proliferation and fiber differentiation by modulating ERK1/2 activity, both in vivo and in vitro [7,8,24]. Cells in lens epithelial explants transduced to overexpress Spred-supressed growth factor-induced ERK1/2 phosphorylation, blocking lens fiber cell differentiation [24]. Furthermore, transgenic mice lines with elevated levels of Spreds in the lens demonstrated reduced levels of ERK1/2 phosphorylation and a concomitant decrease in the rate of lens cell proliferation and fiber differentiation, resulting in a smaller lens phenotype [7]. Interestingly, the small lens phenotype was also observed in transgenic mice that were deficient in Spred1 and Spred2 [8]. Reports by Yoshimura of mice deficient for all alleles of *Spred1* and *Spred2* were shown to be embryonic lethal, resulting from subcutaneous oedema and hemorrhage [6]. These mutants displayed impaired eye size, with the small eye phenotype/microphthalmia characterized during early embryogenesis by our laboratory [8], demonstrating irregular lens epithelial and fiber cell activity. Neither one of the Spreds nor their combination appeared to be required for early lens induction leading to the formation of the primary lens fiber cells; however, Spreds were required for early stages of lens growth [8].

A limitation of the germline knockout model deficient for both *Spred1* and *Spred2* was its embryonic lethality at early stages of lens development [6,8]. To circumvent this embryonic lethality, here, we adopted a conditional knockout approach using Cre-lox recombination to specifically delete both *Spred1* and *Spred2* expression in the lens [25], allowing us to better characterize ocular development at later stages of embryogenesis, fetal development, and postnatal growth. We confirm that Spreds are primarily essential for tightly regulating lens epithelial proliferation and fiber differentiation at early stages of lens morphogenesis, with later stages of lens growth during fetal development and postnatally being less dependent on Spreds. This later reduced dependence of lens growth on Spred proteins is consistent with other related RTK antagonists, such as Sprouty, playing a more integral, compensatory role beyond embryogenesis.

## 2. Materials and Methods

### 2.1. Animals

All procedures using animals were carried out in accordance with the guidelines of the Association for Research in Vision and Ophthalmology and the National Health Medical Research Council (Australia). All procedures were approved by the Animal Ethics Committee, The University of Sydney (NSW, Australia). *Spred1* floxed mice and *Spred2* germline knockout mice were provided by Professor Akihiro Yoshimura (Keio University School of Medicine, Japan) [3]. The MLR10-Cre transgenic mice line was provided by Professor Michael Robinson (Miami University, USA) [25].

#### 2.1.1. Mating and Embryonic Tissue Collection

Mature female mice were injected intra-peritoneally with 5IU of Folligon^®^ (pregnant mare serum gonadotrophin; Lyppard, Sydney, Australia) diluted with 0.9 M NaCl. Forty-seven hours later, just before mating, they were administered a subsequent injection of 5IU of Chorulon^®^ (human chorionic gonadotrophin, Lyppard) diluted in 0.9 M NaCl. Midnight was registered as conception, and the embryos were collected at noon of the post-conception day (namely, E12.5, E14.5, and E16.5). Embryonic head tissue was collected for histology, and limb, tail, or ear tissue was collected for genotyping. 

#### 2.1.2. PCR Genotyping

The following primers and thermal cycling conditions were used for PCR of *Spred-1*flox (Spred1F): 5′-aca act tgc tgg ctg aat ac caca g-3′ and 5′-ctt ctt agt tct gag gta gat gtg c-3′. Initial denaturation occurred at 94 °C for 2 min, denaturation at 95 °C for 15 s, annealing at 55 °C for 15 s, and extension at 72 °C for 10 s (all for 40 cycles), with the final extension at 72 °C for 10 s.

For *Spred2*, the following primers and PCR conditions were used: 5′-gta gcc cag gct ggc aga gaa ctc ac-3′; 5′-gcg cac gta gga gtc cgc gtc ttc-3′; and Neo/Spred2−/−, 5′-cga gat cag cag cct ctg ttc cac atc c-3′, with the initial denaturation at 95 °C for 3 min, denaturation at 95 °C for 15 s, annealing at 60 °C for 15 s, and extension at 72 °C for 10 s (all repeated for 30 cycles), with a final extension at 72 °C for 10 s [3].

The following primer sets and cycling conditions were used for Cre-recombinase: 5′ cct gtt ttg cac gtt cac cg-3′ and 5′-atg ctt ctg tcc gtt tgc cg-3′, with denaturation at 94 °C for 3 min, denaturation at 94 °C for 30 s, annealing at 57 °C for 30 s, and extension at 72 °C for 10 s (repeated for 35 cycles), with the last extension at 72 °C for 2 min. 

#### 2.1.3. Histology 

All tissues collected for histology were fixed in 10% neutral-buffered formalin (NBF) and processed for embedding in paraffin. Seven-micrometer-thick mid-sagittal sections were used for immunofluorescence (Spred1, Spred2, β-crystallin, γ-crystallin, pERK1/2, Sprouty1, and Sprouty2) or periodic acid-Schiff (PAS) staining, as previously described [7,8,26]. Slides were viewed and photographed using brightfield and epi-fluorescence microscopy (Leica-DMLB, Mannheim, Germany). 

### 2.2. Cell Proliferation

#### 2.2.1. BrdU Incorporation

For proliferation studies, pregnant dams were intraperitoneally injected with 5-bromo-2-deoxyuridine (BrdU; Sigma-Aldrich, Burlington, MA, USA, 1 μg per 10 g of body weight) one or two hours prior to tissue collection. For the BrdU-tracer study [27], pregnant mice were injected intra-peritoneally with BrdU 48 h prior to embryo collection. Collected tissues underwent histological processing as previously described [7]. Tissue sections were deparaffinized and hydrated to 50% ethanol, before treatment with 3% H_2_O_2_ in 10% methanol in PBS (*v*/*v*) to block endogenous peroxidase activity. Tissue sections were then digested with 0.02% pepsin in 0.01 M HCL for 20 min at 37 °C, before treatment with 2 M HCL for 30 min, followed by a rinse in PBS and deionized H_2_O. Sections were treated with 3% normal goat serum (NGS, Sigma-Aldrich) for 60 min prior to incubating overnight at 4 °C with an antibody against BrdU (DAKO, Glostrup, Denmark; diluted 100-fold with 3% NGS). The following day, sections were incubated with a secondary anti-rabbit IgG Alexa Fluor 594 antibody and diluted 1000-fold in PBS for 2 h at room temperature, followed by a rinse with PBS. For the nuclear counterstain, slides were incubated for 5 min with 1 mg/mL of Hoechst dye (bisbenzimide, Calbiochem; San Diego, CA, USA). Slides were then washed with PBS and mounted with 10% PBS in glycerol. BrdU-positive cells were counted as a percentage of the total number of epithelia to determine the rate of cell proliferation. In order to increase accuracy, each count was the average taken from three consecutive sections from at least three slides, from at least three different lenses.

#### 2.2.2. EdU Incorporation

For EdU incorporation, pregnant mice were injected intra-peritoneally with EdU 1 h prior to collection of embryos. EdU was diluted in sterile 0.9 M NaCl and administered at 1 mg per 10 g of body weight. Collected tissues underwent histological processing, and sections were de-paraffinized and hydrated as previously described [7]. The tissue was treated with 1% Tween in PBS for 20 min, before using the Click-iT EdU Alexa Fluor 488 Imaging kit (#c10337 Invitrogen; Mount Waverley, Australia) to detect EdU incorporation. The Click-iT reaction cocktail was applied to the slides for 30 min, then washed off in PBS. For the nuclear counterstain, the slides were incubated for 5 min with 1 mg/mL of Hoechst dye (Sigma-Aldrich). Slides were rinsed in PBS and mounted using glycerol with 10% PBS. 

### 2.3. Lens Epithelial Explants

All collection, dissection, and subsequent culture of tissues was conducted in Medium (M199; pH 7.2; Thermo Fischer Scientific; Nedlands, Australia) with Earle’s salts, supplemented with 2.5 μg/mL amphotericin B (Sigma-Aldrich), 50 μg/mL penicillin/streptomycin (Thermo Fisher Scientific), 0.1 μg/mL glutamine (Thermo Fischer Scientific), and 0.1% albumen from bovine serum (BSA, Sigma-Aldrich). P10 mice eye lenses were obtained to prepare lens epithelial explants as previously described [28]. Human recombinant fibroblast growth factor 2 (FGF2; 50 μg; Preprotech, Cranbury, NJ, USA) was reconstituted as per the manufacturer’s instructions and used at a concentration of 5 ng/mL for proliferation assays, or 100 ng/mL for fiber differentiation assays. FGF2 was administered to explants, and cells were cultured with for 24 h (proliferation assay) or 10 days (differentiation assay), prior to being fixed for 20 min with 10% NBF for subsequent immunofluorescent labeling. 

#### Immunofluorescent Labeling

Fixed explants were permeabilized in PBS/Tween 20 (Sigma-Aldrich) and were treated with 3% (*v*/*v*) NGS. Primary antibodies (anti-β-crystallin 1:50, E-cadherin 1:100, pERK1/2 1:100, and BrdU 1:100) were applied to the explants and left overnight at 4 °C in a humidified chamber. Explants were then rinsed with PBS and incubated with the appropriate secondary antibody (Alexa Fluor 594 or 488, diluted in 1:1000) for 2 h at room temperature, before being counterstained with Hoechst dye for 5 min. Explants were mounted with PBS/glycerol and were viewed and photographed using epi-fluorescence microscopy (Leica-DMLB, Germany). 

### 2.4. Analysis 

For all measurements, mid-sagittal sections of the eye (at least 3 consecutive sections per slide per embryo) were used, where ‘*n*’ represents the number of lenses (from at least 3–5 different embryos for each experiment). To quantify lens fiber cell length, lens measurements were taken from the apical side of the lens epithelium to the posterior fiber pole. The total cross-sectional area confined by the lens capsule of mid-sagittal sections was used to measure the lens area. 

To determine the number of epithelial cells, the epithelial cells from the transitional region fulcrum were counted from one side of the lens section to the other. We defined the central and peripheral regions of the epithelium using the placement of the tip of the iris as the boundary between the two zones. Data are expressed as +\− the standard error of the mean (SEM). 

All experiments were repeated at least in triplicate, using at least 3 different embryos at each gestation period. All images for immunolabeling were photographed with the same exposure settings, and the same brightness and contrast settings were applied to all images to highlight the respective differences. Immunofluorescence of the lens region (area confined by the lens capsule in mid-sagittal sections) was quantified, where the relative intensity was measured using ImageJ software (version 6.1.0). Statistical analysis was conducted using non-parametric Mann–Whitney tests, performed to determine the data’s significance (* *p* < 0.05).

## 3. Results

### 3.1. Conditional Knockout of Spred1 and Spred2 in the Murine Lens

Embryonic and postnatal mice were genotyped to identify the presence of the floxed allele for Spred1 and Cre-recombinase, together with the loss of the Spred2 alleles (Figure 1). For Spred1, the following PCR products were observed: the WT allele at 476 bp (Figure 1A), the floxed (null) allele at 511 bp (Figure 1A), with bands for both WT and floxed alleles in heterozygous mice (Figure 1A). For Spred2, the PCR products corresponded to the WT allele at 600 bp (Figure 1A) and the null allele at <400 bp (Figure 1A), with heterozygous mice (Figure 1A) showing both bands for the WT and null alleles. A Cre-recombinase-positive band was observed at 300 bp (Figure 1A), with the absence of a band signifying Cre-recombinase-negative lines (Figure 1A). Conditional knockout (CKO) lines deficient for both Spred1 and Spred2 (Spred1/2^CKO^) in the lens showed a 511 bp band for Spred1, a <400 bp band for Spred2, and positive band for Cre-recombinase at 300 bp. For this study, mice heterozygous for the null alleles for Spred1 and/or Spred2 were used as some age-matched littermate controls for histological analysis, given they did not present any lens/eye abnormalities when compared to WT mice. These were all collectively referred to as control tissue.

E14.5 embryos were employed to compare the levels of Spred protein immunoreactivity in the lens of control (Figure 1B–E) and Spred1/2^CKO^ (Figure 1F,G) mice. Spred1 protein was detected throughout the whole lens in control tissue (Figure 1C), with stronger labeling in the lens epithelium and primary fiber cells. Spred2 labeling was most prominent in the fiber cells of the control lens (Figure 1E). In contrast, the Spred1/2^CKO^ lens failed to label for either of the Spred proteins (Figure 1G).

### 3.2. The Loss of Spred1/2 in Lens Results in Microphakia

When comparing lens morphology of Spred1/2^CKO^ mice to that of control mice at different ages (E12.5 to P1; Figure 2), we observed microphakia, with a smaller lens most prominent at earlier stages of embryogenesis. At E12.5 and E14.5, the control lens (Figure 2A,C) displayed its distinct lens polarity and architecture, with the primary fiber cells in contact with the overlying epithelial monolayer. In contrast, lenses deficient for Spred1 and Spred2 (Figure 2B,D) had an aberrant morphology, with distinctly shorter fiber cells still in contact with the epithelium, contributing to the small lens phenotype. There were no notable differences in the lens capsule, with a similar thickness displayed in both Spred1/2^CKO^ and control lenses. Accompanying the microphakia at E12.5, when compared to control eyes, there was an equally smaller developing retina/optic cup in the Spred1/2^CKO^ lenses, highlighting microphthalmia in early ocular morphogenesis (Figure 2A,B).

To characterize the small lens phenotype, we quantified the average length of the lens fibers by taking apical to basal measurements of mid-sagittal sections of the lens. Spred1/2^CKO^ lenses displayed significantly shorter fiber cells at E12.5 (Figure 2I), with the average fiber cell length recorded as 657 AU ± 36.7 (*n* = 7, *p* = 0.0012) compared to control lens 856 AU ± 34.49 (*n* = 7), a reduction of ~23%. This was also the case, though not as severe, at E14.5 (Figure 2K), with a ~14% reduction in fiber cell length, with the average length in Spred1/2^CKO^ lenses at 1088 AU ± 27.71 (*n* = 8, 0.0013) compared to that of the control lens at 1264 AU ± 28.01 (*n* = 6). The total cross-sectional area confined by the lens capsule of mid-sagittal sections was also collated and quantified, being more severe earlier in development; the total lens area at E12.5 (Figure 2J) and E14.5 (Figure 2L) of Spred1/2^CKO^ mice was significantly smaller at 32.06 µm^2^ ± 2.50 (*n* = 7, *p* = 0.0025) and 78.21 μm^2^ ± 5.207 (*n* = 6, *p* = 0.0022), respectively, when compared to the control lens at 61.31 µm^2^ ± 3.17 (*n* = 5) and 121.6 μm^2^ ± 3.91 (*n* = 6), respectively (Figure 2J,L). 

Interestingly, the small lens phenotype associated with the loss of Spred1 and Spred2 became less evident with age, appearing to recover in size between the ages of E14.5 and E16.5. By E16.5, lens morphology (Figure 2E,F), fiber cell length (Figure 2M), and lens area (Figure 2N) no longer demonstrated any significant difference between control and Spred1/2^CKO^ mice. When examined postnatally, the Spred1/2^CKO^ lens at P1 was similar to the intact, symmetrical, biconvex lens of control mice (Figure 2G,H). In some instances, the Spred1/2^CKO^ lens appeared to closely adhere to/associate with the overlying cornea, resulting in an irregular anterior lens morphology. Overall, postnatally, there were no significant differences in fiber cell length or lens area (Figure 2O,P). Notably, as the Spred1/2-deficient mice lenses recovered in size with age, so did the delayed development of the surrounding ocular tissue, namely, the optic cup/retina.

### 3.3. Increased Epithelial Cell Population in Spred1/2-Deficient Lenses

When examining the epithelium of the Spred-deficient lenses, there was an evident disruption to its native monolayer, with epithelial cells appearing multi-layered/pseudostratified throughout development (Figure 3). Although this lens epithelial phenotype was more pronounced at earlier developmental stages, i.e., E12.5 (Figure 3B) and E14.5 (Figure 3D), it still remained apparent at E16.5 (Figure 3F) and P1 (Figure 3H). Unlike the recovery of the small lens size with age, the lens epithelial phenotype persisted, correlating with a significant increase in epithelial cell density. 

Quantitative analysis of the average epithelial cell numbers in mid-sagittal sections of Spred1/2^CKO^ lenses showed significantly higher numbers than that of control lenses at all ages examined: at E12.5, a ~32% increase (Figure 3I): control 58.50 ± 3.37 (*n* = 6) vs. Spred1/2^CKO^ 85.67 ± 2.94 (*n* = 6, *p* = 0.0079); at E14.5 (Figure 3J): control 89.17 ± 3.03 (*n* = 6) vs. Spred1/2^CKO^ 119.3 ± 4.23 (*n* = 10, *p* = 0.0020); at E16.5 (Figure 3K): control 131.2 ± 6.85 (*n* = 6) vs. Spred1/2^CKO^ 186.7 ± 4.43 (*n* = 6, *p* = 0.0051); and at P1, a ~14% increase (Figure 3L): control 177.6 ± 3.28 (*n* = 5) vs. Spred1/2^CKO^ 204 ± 7.75 (*n* = 5, *p* = 0.0079). This increase in epithelial cell numbers was also evident and validated at P10, when examining ex vivo lens epithelial wholemounts. To account for the differences in epithelial cell numbers, we assessed lens tissue for differences in rates of epithelial cell proliferation. 

### 3.4. Spred1/2-Deficient Lenses Have Increased Rates of Epithelial Cell Proliferation

Cell proliferation was examined at both embryonic and postnatal ages in intact Spred1/2^CKO^ and control lenses using immunolabeling for BrdU-incorporation in tissue exposed for either 1 h (Figure 4) or 2 h. The 2 h exposure showed a significant increase in the number of BrdU-labeled (proliferating) epithelial cells in the Spred1/2^CKO^ lenses compared to the control lens; however, when related to the increased cell numbers, there was no significant difference in the overall rate of cell proliferation between the Spred1/2^CKO^ and control lenses. When repeated with a more stringent 1 h window of exposure to BrdU, there was a significant increase in the rate of epithelial cell proliferation at E14.5 in Spred1/2^CKO^ lenses (Figure 4B,E), at 22.2% ± 2.81 (*n* = 9; *p* = 0.0212) compared to control lenses (Figure 4A,E) at 14.1% ± 2.07 (*n* = 12). A similar increase in the rate of epithelial cell proliferation was also seen postnatally at P1: Spred1/2^CKO^ lenses (Figure 4D,F) were at 21% ± 3.0 (*n* = 2), compared to the control lens (Figure 4C,F) at 10% ± 0.47 (*n* = 4). 

To determine whether the increased rate of epithelial cell proliferation was a direct result of the loss of Spred1 and 2, we labeled for BrdU in epithelial explants derived from mutant mice deficient for Spred1/2 stimulated with a proliferative dose of FGF2 (5 ng/mL) for 24 h. In control lens epithelial explants not treated with FGF2 (Figure 5A), the proliferation rate was 5.6% ± 0.39 (*n* = 3). With the addition of FGF2 (Figure 5C), this rate of proliferation was significantly increased to 25.4% ± 0.22 (*n* = 3). When compared to the Spred1/2^CKO^-derived lens epithelial explants, the basal rate of cell proliferation without FGF2 (Figure 5B) was 7.6% ± 1.8 (*n* = 3), greater than but not significantly different from that seen in the control tissue without FGF2. With the addition of FGF2 to the Spred1/2^CKO^ lens cells (Figure 5D), there was an increased rate of proliferation at 38.8% ± 4.4 (*n* = 3). Statistical analysis (Figure 5E) revealed cell proliferation was significantly increased in control lens cells with and without the addition of FGF2 (*p* = 0.0017 **), between the Spred1/2^CKO^ mice, with and without FGF addition (*p* ≤ 0.0001 ****), as well as between the control and Spred1/2^CKO^ lens explants, both treated with FGF (*p* = 0.0170 *).

Given the aberrant rate of epithelial cell proliferation observed in Spred1- and 2-deficient lenses, we questioned whether the number of lens fiber cells and/or the rate of epithelial cell differentiation into secondary lens fibers was also compromised.

### 3.5. Loss of Spred1/2 in the Lens Impacts the Rate of Fiber Cell Differentiation

We initially characterized for in situ changes in lens fiber differentiation of Spred1/2^CKO^ lenses by examining the rate of epithelial to fiber cell differentiation (tracking BrdU-labeled epithelial cells for 2 days from E12.5 to E14.5 (E12.5 + 2)). Proliferating epithelial cells (at E12.5) incorporate BrdU within the first 2 h of exposure, and when left for a subsequent 2-day growth period (E12.5 + 2 = E14.5), we can follow the fate of these BrdU-labeled cells [27]. For example, BrdU-labeled cells observed in the lens cortex are the result of migrating BrdU-tagged cells subsequently differentiating into secondary fibers over the 2-day window. Using this ‘BrdU-tracer’ assay, we counted and quantified the number of ‘tagged’ lens epithelial cells entering the lens cortex as secondary fibers between control and Spred1/2^CKO^ mice.

It was revealed that after 2 days, the Spred1/2^CKO^ lenses (Figure 6B) had an increased rate of fiber cell differentiation, with significantly more BrdU-positive fiber cells in the lens cortical region (18.5% ± 1.69, *n* = 5, *p* = 0.0173) compared to control lenses (Figure 6A; 9.6% ± 2.03, *n* = 6). The extent to which these epithelial cells were displaced into the lens cortex as fiber cells was also noted, with labeled fiber cells of Spred1/2^CKO^ lenses positioned deeper into the cortical region within the 48 h time tracing period (Figure 6, arrows), supportive of an increased migration and differentiation rate of secondary fiber cells. There was also a notable spatial difference of BrdU-labeled cells between the Spred1/2^CKO^ and control lenses. The control lens had a relatively equal distribution of BrdU-labeled cells between the lens epithelium and fiber cell region, whereas in the Spred1/2^CKO^ lenses, the majority of the BrdU-tagged cells after 48 h appeared to be fibers (Figure 6B, arrows). When quantified (Figure 6E), the percentage of BrdU-labeled epithelial cells after 48 h significantly (*p* = 0.04) decreased, from 10.8% ± 0.70 (*n* = 6) in the control lens to 5.86% ± 1.49 (*n* = 7) in the Spred1/2^CKO^ lens. 

The intensity of the BrdU label in the nuclei of cells provided additional insights into lens epithelial cell behavior, in particular the rate of individual cell replication. In the Spred1/2^CKO^ lens (Figure 6B), epithelial cells tagged with BrdU that went on to immediately differentiate into fibers (white arrows) maintained a relatively strong BrdU label in their nuclei (highlighted in Figure 6G). The BrdU-tagged cells in the epithelium of Spred1/2^CKO^ lenses had a markedly weaker (‘more dilute’) BrdU label (green arrows; highlighted in Figure 6I), indicating multiple cell divisions within the 2-day tracer period. In contrast, in the control lens (Figure 6A), we see a consistent intense label for BrdU, both in the epithelial and fiber cell regions (yellow arrow; highlighted in Figure 6H). These findings suggested that the rate of individual cell division in the Spred1/2^CKO^ lens epithelium was greater than that observed in the control lens. 

To validate that there were differences in the rate of lens epithelial cell proliferation in the absence of Spred1/2, we co-labeled our tracer study tissues with EdU, administered 1 h prior to the collection of the E14.5 (E12.5 + 2) embryos. The number of actively proliferating epithelial cells that labeled with EdU (green nuclei) displayed the same significant increase as observed earlier (see Figure 4) in both control (Figure 6C,E) and Spred1/2^CKO^ lenses (Figure 6D,F).

### 3.6. Loss of Spred1/2 in Lens Impacts Fiber Cell Numbers

Given the significant increase in the rate of lens fiber differentiation observed in the Spred1/2^CKO^ lens, we compared the number of fiber cells (based on number of fiber cell nuclei) in mid-saggital sections of Spred1/2^CKO^ and control lenses. At E16.5, the lens cortex had a higher fiber cell density in the Spred1/2^CKO^ lens (Figure 7B) compared to the control lens (Figure 7A), particularly in the bow zone, where new secondary fibers form (Figure 7B, arrows). The number of fiber cells in the Spred1/2^CKO^ lenses was significantly higher at E12.5 (Figure 7E; 158 ± 2.71, *n* = 6, *p* = 0.0022), E14.5 (Figure 7F; 203 ± 10.42, *n* = 9, *p* = 0.0028), and E16.5 (Figure 7G; 303 ± 1.095, *n* = 6, *p* = 0.0022) when compared to the respective control lens, 112 ± 6.49 (*n* = 6), 149 ± 7.92 (; *n* = 6) and 184 ± 8.258 (*n* = 6). Postnatally, the fiber cell region once again appears to have a higher cell density in the Spred1/2^CKO^ lens (Figure 7D) at P1 when compared to the control lens (Figure 7C); however, when quantified (Figure 7H), there was no significant difference in fiber cell numbers between control (225 ± 30.25, *n* = 9) and Spred1/2^CKO^ (292.7 ± 50.71, *n* = 7) lenses. It should be noted that there was no indication of cell death in the mutant lenses at any stage. 

### 3.7. Loss of Spred1/2 in Lens Impairs Crystallin Expression

The accumulation of β- and γ-crystallins is a hallmark feature of lens fiber cell differentiation, and this was examined in both Spred1/2^CKO^ and control tissue throughout lens development at E12.5, E14.5, E16.5, and P1 (Figure 8). β-crystallin labeling in the Spred1/2^CKO^ lens (Figure 8D,F,H) was very similar to that of the control lens (Figure 8C,E,G) from E14.5 to P1, with no notable differences in label; however, at E12.5, β-crystallin levels appeared relatively stronger in the newly formed fiber cell region and lens epithelium of the Spred1/2^CKO^ lens (Figure 8B) when compared to the aged-matched control lens (Figure 8A). When examining γ-crystallin labeling, it appeared earlier in the fiber differentiation process, with a relatively stronger label in the Spred1/2^CKO^ lens (Figure 8J,L,N,P) compared to the control lens (Figure 8I,K,M,O). From E12.5 (Figure 8J), the onset of γ-crystallin also appeared gestationaly earlier in the Spred1/2^CKO^ lens fibers compared to the control lens (Figure 8I). This trend persisted with increasing age (Figure 8). 

### 3.8. Loss of Spred1/2 in Lens Cells Impairs FGF-Induced Lens Fiber Cell Differentiation

To validate our in situ findings, we prepared lens epithelial explants from P10 Spred1/2^CKO^ and control mice and compared their ability to undergo lens fiber cell differentiation in response to FGF2. FGF2-treated lens explants showed strong labeling for β-crystallin in both control (Figure 9E) and Spred1/2^CKO^ (Figure 9F) lens cells when compared to explants not treated with FGF2 (Figure 9D). β-crystallin labeling was present in many of the Spred1/2^CKO^ (Figure 9F) cells treated with FGF. These were not as elongated when compared to the differentiating fibers in FGF-treated cells of control explants (Figure 9E). The membranous labeling of E-cadherin in the epithelia of non-treated explants (Figure 9G) was lost in the FGF2 treatment groups, confirming the loss of the epithelial phenotype as the cells underwent fiber cell differentiation. Other than the resultant differentiating fibers induced by FGF in Spred1/2^CKO^ explants appearing less elongated, we did not observe any noticeable differences in the onset or the rate of fiber cell differentiation between the mutant and control lens cells.

### 3.9. Spreds Negatively Regulate Phosphorylation of ERK1/2 in Lens Cells

As Spred proteins are reported to directly regulate the ERK1/2-signaling pathway, we examined and compared the phosphorylation state of ERK1/2 in lens cells deficient for Spred1 and Spred2. Immunolabeling for phosphorylated ERK1/2 (pERK1/2) was elevated and relatively stronger in the lenses of Spred1/2^CKO^ mice compared to control lenses throughout development (Figure 10). Spred1/2^CKO^ lenses showed increased levels of pERK, particularly in the germinative and transitional lens regions at E12.5 (Figure 10B) and E14.5 (Figure 10D), when compared to the control lens (Figure 10A,C). At E16.5 and P1, compared to the control lens (Figure 10E,G), the Spred1/2^CKO^ lens (Figure 10F,H) continued to display an increased level of pERK1/2 labeling, with a prominent label in the transitional zone and a robust label in the newly forming secondary fiber cells. This phenomenon was also replicated in P10 lens epithelial wholemounts, demonstrating a significantly (*p* = 0.0079) elevated level of pERK labeling in the untreated Spred1/2^CKO^ cells (Figure 10J; 36.71 ± 2.42 AU, *n* = 5) compared to control lens cells (Figure 10I; 20.30 ± 1.92 AU, *n* = 5).

### 3.10. Sprouty1/2 Levels Increase in the Absence of Spred1/2 in the Lens

We investigated other related ERK/MAPK antagonists found in the lens, namely, Sprouty1 and Sprouty2, to determine whether there was any putative compensatory relationship with the absence of Spreds, accounting for the postnatal recovery of the mutant lens phenotype. Immunoreactivity for Sprouty1 (Figure 11A–F) was seen throughout the whole lens. The Spred1/2^CKO^ lens showed a marked increase in levels of Sprouty1, particularly in the fiber cells at E12.5 (Figure 11B) and E14.5 (Figure 11D), compared to the control lens (Figure 11A,C). At E16.5, the levels of Sprouty1 in the Spred1/2^CKO^ lens (Figure 11F) appeared to be similar and more consistent with the control lens (Figure 11E). Sprouty2 immunoreactivity, like that for Sprouty1, was also observed throughout the whole lens (Figure 11). The Spred1/2^CKO^ lens showed markedly increased levels of Sprouty2, particularly in the epithelium at E12.5 (Figure 11H) and E14.5 (Figure 11J), when compared to the control lens (Figure 11G,I). By E16.5, the levels of Sprouty2 in the Spred1/2^CKO^ lens (Figure 11L) appeared to be relatively increased in the fiber cell region when compared to the control lens (Figure 11K).

## 4. Discussion

This study examined the specific role of Spred1 and Spred2 during lens development and growth. Our previous studies characterizing germline mutant mice deficient in Spred1 and/or Spred2 highlighted that the Spred proteins were not essential for early lens induction or the formation of primary lens fiber cells; however, they were found to play an important role in the growth of the embryonic lens [8]. In the absence of Spreds, lens and eye size were significantly compromised, with evidence of aberrant lens epithelial cell proliferation and fiber cell differentiation, which was associated with increased ERK1/2 activity in the lens [8].

The greatest limitation of using the germline knockout mice was that the loss of Spreds conferred embryonic lethality at earlier stages of lens development; hence, we could not effectively fully follow and characterize the eye phenotype. Given the mice had microphthalmia, it was not clear whether the small lens phenotype was intrinsic to the lens or the result of microphthalmia, suggesting Spreds could be acting more widely. In the present study, we introduced a conditional knockout approach to circumvent embryonic lethality and to fully characterize ocular lens development in the absence of Spred1 and Spred2 specifically in the lens.

### 4.1. Conditional Loss of Spreds in Lens Results in Microphakia and Microphthalmia

Mice with lenses specifically deficient for Spred1 and Spred2 displayed microphakia, confirming the loss of Spreds directly impacts the lens and is not the result of extrinsic factors associated with germline deletion of both Spreds, as shown previously [8]. Interestingly, though the severity of the small lens phenotype of CKO mice was comparable to that seen in germline knockout lines, there was a subsequent temporal recovery of lens size. Using the MLR10-Cre line, we observed a specific deletion of Spreds at the lens vesicle stage, from E10.5, and by E16.5, we had noticed a significant recovery in lens size, comparable to control lenses. The small lens phenotype compromised whole eye development, further highlighting a key role for the lens in regulating eye growth and development. The impact on eye growth was shown in the double Spred CKO lines, as well as when Spred1 alone was lost, but not in the absence of only Spred2 from lenses. 

Earlier studies reported that with retardation of lens growth, this can lead to microphthalmia, and in more severe cases, where there is aphakia (no lens formed), the surrounding eye structures either subsequently degrade or do not develop [29,30,31]. Our current findings align with these past studies, with the lens influencing the overall growth and shape of the eye. With that said, we cannot rule out the notion that the inherent loss of Spreds can also have a direct impact on the severity of the phenotype observed in the extra-lenticular ocular structures, as it was seen to differ between the germline and tissue-specific knockout lines.

### 4.2. Spred Negatively Regulates Lens Epithelial Cell Proliferation

When the growth—in particular, the size of a tissue—is compromised, it is often attributed to a decline in the rate of proliferation and aberrant differentiation, two key cellular processes required for the development and sustained growth of the lens. In our mutants with microphthalmia, we observed the opposite. In smaller lenses of the Spred1/2^CKO^ mice, we found a significantly higher rate of lens epithelial cell proliferation throughout embryogenesis and into early postnatal growth, which disrupted the lens epithelial monolayer that remained multilayered. We validated this in vitro, using lens epithelial explants prepared from these mutant Spred1/2^CKO^ mice, exposing them to a proliferating dose of FGF. The rate of FGF-induced cell proliferation in these lens cells was 10 percent greater than that seen with explants prepared from control lenses. This is consistent with the fact that when Spred is overexpressed in lens epithelial cells, there is an opposite, significant decrease in FGF-induced proliferation and cell numbers [7], highlighting an important endogenous regulatory role for Spreds in lens cell proliferation. Spred’s antagonism of lens cell proliferation is consistent with its negative role in other cell types, including hematopoietic cells, keratinocytes, mast cells, and human hepatocellular carcinoma cells [32,33,34]. 

Our findings align with similar studies overexpressing Sprouty in the lens, another RTK antagonist related to Spreds. Elevated levels of Spry in the lens, like Spred, also lead to a reduced number of lens epithelial cells, concomitant with a reduced rate of epithelial cell proliferation [26]. Interestingly, in contrast to the Spred mutants we report here, mice deficient for Spry in the lens do not show any significant changes to their proliferative behavior [26,34]. 

### 4.3. Spred Negatively Regulates Lens Fiber Cell Differentiation (Elongation and Maturation)

The increase in the rate of lens cell proliferation we observed correlated with an increase in the rate of fiber cell differentiation, verified using an in vivo tracer assay to follow proliferating lens epithelial cells (see [27]). Over a 48 h period of rapid lens growth, we monitored the rate of Spred1/2^CKO^ mutant lens epithelial cells differentiating into secondary fiber cells. The Spred1/2^CKO^ mutant lenses had significantly more labeled fiber cells in the lens cortical region after the 48 h period compared to control lenses, indicating an increased rate of fiber differentiation. These cells had also migrated further, deeper into the cortex. Similarly, as we eluded to earlier for cell proliferation, we see a converse result when Spred is overexpressed in lens cells, with impaired epithelial to secondary fiber cell differentiation, as cells were retained in the transitional zone [7]. This trend also extends to other RTK antagonists, such as Spry, with Spry2 overexpression in the lens demonstrating a similar significant decrease in the number of labeled fiber cells moving into the lens cortex [35]. Interestingly, using a novel co-tag of lens epithelia with BrdU and subsequently EdU, we found that in the Spred mutant lens, more epithelial cells were undergoing fiber differentiation at the expense of epithelial cell renewal, with control lenses having a more equitable share of new epithelial cells and fiber cells.

With the increased rate of fiber cell differentiation, we also observed notable differences in β-crystallin labeling of the mutant lenses at E12.5 when compared to control lens, with stronger labeling in the newly forming fibers. This was validated by the earlier onset of γ-crystallin labeling in mutant lenses at E12.5 and throughout development, suggesting that the heightened rate of fiber differentiation was accompanied by an earlier primary and secondary fiber cell maturation. Further studies investigating other molecular markers of fiber cell differentiation and maturation and cell cycle regulators will provide a greater insight into the regulatory ability of Spreds. Moreover, investigations into the morphological changes associated with fiber cell maturation, including the onset of autophagy, would be very informative, particularly during rapid lens growth at E16–18, where mice begin to lose their fiber cell nuclei. 

Together, we see that the combined phenotype in the lens epithelium and fiber cells contributes to a smaller lens. Paradoxically, despite the heightened rate of lens cell proliferation and the differentiation into fiber cells, lens growth is temporally stalled. This may be due to the fact that the hyperproliferative lens epithelia are mostly retained and become multilayered, in place of an apical monolayer expanding over the fiber mass. Whether Spreds play a role in maintaining the epithelial apical basal polarity for the cells to become multilayered warrants further investigation. Moreover, though the rate of secondary fiber differentiation is also pronounced in these mutants, the fact that fiber cell length—and most likely their volume—is compromised, limits the height and area of the lens, impacting its size. Given the differential expression of Spreds in the lens, it is still unclear whether these two cellular defects (epithelial and fiber) are mutually exclusive or are interconnected to result in microphakia. The single knockout lens phenotypes of Spred1 and Spred2 would suggest the latter. We should also not rule out the differences in the regulation of the lens cellular processes during embryogenesis vs. postnatally. 

### 4.4. Spred Negatively Regulates ERK1/2 Signaling 

Given the important role of Spred proteins as specific regulators of MAPK/ERK1/2 signaling [1,32,36], modulating cell behavior, it was important to evaluate the activity of ERK1/2 in our mutant lenses. Earlier studies have linked the importance of the onset, duration, and levels of ERK1/2 activity to determining cell fate and/or the rate of cellular processes, such as lens epithelial cell proliferation and fiber cell differentiation [7,15,17,18]. We report significantly increased pERK1/2 levels in the lens in the absence of both Spred1 and Spred2, especially at early stages of development. The elevated levels of pERK1/2 were primarily observed in the germinative, transitional, and newly forming fiber cell regions of the lens when compared to the control lens. This heightened activity of pERK1/2 most likely contributed to the elevated levels of lens cell proliferation and differentiation. 

Ex vivo whole mounts of lens epithelia highlighted how effective Spreds are in suppressing ERK1/2 phosphorylation, as lens cells isolated from mutants showed much stronger endogenous ERK1/2 activity (phosphorylation) compared to equivalent tissue from control lenses. Unlike the in vivo studies, there were no significant changes in the localization of pERK1/2 in mutant mice lens epithelium when compared to the control. This difference may be attributed to the fact that the in vivo study used embryonic tissue, whereas the in vitro study used postnatal tissue. 

Our findings are consistent with other studies examining Spry, another antagonist shown to effectively inhibit the phosphorylation of ERK1/2 [37]. Disrupting Spry signaling can be detrimental to ERK signaling, with the loss of both Spry1 and Spry2 modulating ERK activation, with increased levels of its phosphorylation noted in Spry-deficient lenses [34,35]. In contrast, elevated levels of Spry can mimic the functional loss of RTKs [37]. Overexpression of Spry2 in lens cells was more active in suppressing ERK1/2 phosphorylation, with a subsequent block to cell proliferation, than overexpression of Spry1 [7]. Conditional loss of Mapk1 (ERK2) in the lens supported this finding, with a reduction in epithelial cell proliferation and cell number embryonically [38]. Collectively, this is consistent with Spred and Spry specifically targeting ERK1/2 signaling [1], which is largely responsible for regulating lens epithelial cell proliferation [16] and maintaining the normal lens epithelial cell population.

Our current findings perfectly complement our earlier studies examining the impact of elevated Spred in the lens, resulting in decreased levels of pERK1/2 and a subsequent reduction in lens cell proliferation and fiber differentiation [7]. Most interestingly, whether Spred is lost or overexpressed in the lens, both scenarios result in a small lens phenotype via two very different mechanisms. Although there may not be a direct link between an increased rate of proliferation and differentiation to the overall size of the lens, any modification of ERK activity clearly leads to aberrant signaling and communication between lens cells, which is required for maintaining the correct size and structure of the lens, emphasizing the importance of finely balanced cell signaling and its different positive and negative regulators to ensure a defined physiological response.

### 4.5. Recovery of Spred-Deficient Embryonic Lens Size

One of the more interesting and novel findings to come from the current study was the progressive recovery in size of mutant mice lenses during fetal growth. Though different factors may have contributed to this phenomenon, we propose that other related antagonists, such as Spry, may compensate for the loss of Spreds later in development, with Spreds less critical for lens growth during the later stages of gestation. 

We previously demonstrated the putative redundancy between different Spred proteins [8], and we now extend this to other related antagonists, such as Spry, especially given that they share a similar C-terminus (Sprouty-like cysteine-rich SPR domain) [39,40], are co-localized in the lens, and both effectively block ERK1/2 signaling [37,41,42,43,44,45]. In the present study, we report that in Spred-deficient lenses, Spry1 levels appear elevated during early development (E12.5, −E14.5); however, by E16.5, they are comparable to control lens levels, coinciding with the recovery of lens size. Spry2, on the other hand, was also elevated at these early stages, but it continued to remain elevated throughout later stages, particularly in the lens epithelium, suggestive of complementary behavior between the Spred and Spry antagonists. Due to the challenge of isolating sufficient protein from lens tissue at early embryonic ages, one of the greatest limitations was the ability to quantify these early protein changes to support our proposed compensation between these antagonists, opening the door for further investigation using transcriptomics and/or alternate tissues and models. 

Spry proteins may play a more significant role later in lens development, and this is supported by previous studies from our laboratory and others, where the lenses of mutant mice deficient for Spry do not show any major phenotype early in development [34,35]. Postnatally, however, the loss of Spry1 and Spry2 in the lens leads to an epithelial to mesenchymal transition (EMT) of lens epithelia and subsequent cataract associated with an aberrant increase in ERK1/2 activity as well as TGFβ/SMAD signaling [35]. Outside of the lens, studies examining Spry expression in the telencephalon of developing mouse embryos showed that Spry deletion had little to no effect on cell behavior at E12.5; however, by E15.5, abnormalities in ERK phosphorylation were detected [46]. This trend is similar to that seen in the lenses of Spry mutants, further supporting the fact that the primary function of Spry may occur later in embryogenesis.

An alternate explanation for the recovery of size in Spred-deficient lenses may relate to the temporal onset of Cre-recombinase using the MLR10-Cre-line to delete Spred1 [25]. Up until E10.5, although the presumptive lens is deficient for Spred2, normal lens induction and development proceeds with only Spred1. In fact, early in murine lens development, Spred1 expression levels are approximately three times greater at E15–E18 than postnatally at P6–P9 [47], implicating that it may be more active in earlier stages of lens development. Taken with our observations, we propose that early lens development and growth may be more dependent on Spred proteins compared to later stages, where related antagonists, such as Spry, and possibly even Sef [48], can effectively compensate for the loss of Spreds we see here. Our current study, combined with findings from previous reports, suggest that Spry and Spred family members tightly regulate the ERK/MAPK pathway, as well as related cellular processes at various stages of development, to maintain lens size and shape, resulting in the observed recovery.

## 5. Conclusions 

Our previous findings using conventional germline KO mouse lines, further substantiated in this study using conditional KO lines, clearly establish that microphthalmia and the small lens phenotype intrinsic to the lens and is the direct result of Spred1/2 loss. Furthermore, Spred1 and Spred2 are instrumental in modulating various cellular processes essential for proper lens development by antagonizing the ERK/MAPK pathway and modulating pERK1/2 levels, known to regulate important cellular processes, such as proliferation and differentiation, to regulate normal lens development and growth. Additionally, we see an eloquent compensatory system between related RTK antagonists, including other homologues of Spred and Spry proteins, where they may play different roles to each other in early embryonic and adult development. Spred proteins may be a putative target for developing novel therapeutic treatments for different pathologies, given Spred is a naturally occurring suppressor of cellular proliferation and differentiation. This extends Spred’s importance well beyond lens development and pathology, sparking the attention of researchers in various cancer and tumorigenesis fields.

## Figures and Tables

**Figure 1 cells-13-00290-f001:**
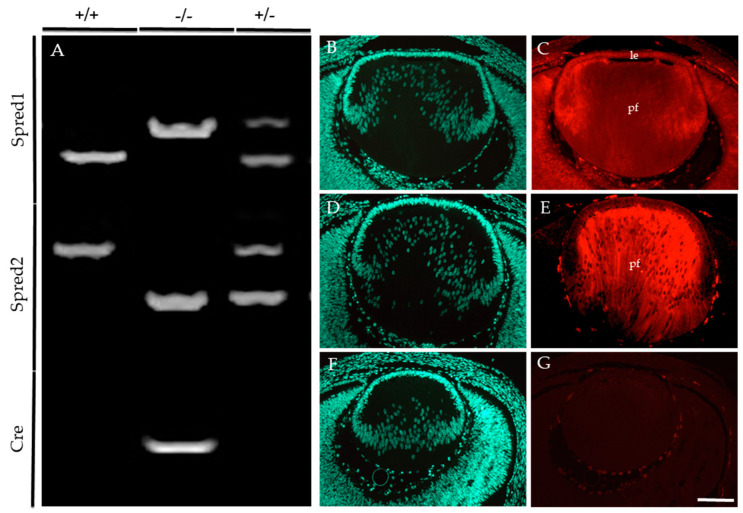
Conditional knockout of Spred1 and Spred2 in the murine lens. The following PCR products (**A**) were expected for Spred1: WT 476 bp, floxed null allele 511 bp, heterozygotes both 476 bp and 511 bp bands. For Spred2, WT 600 bp, null allele < 400 bp, heterozygotes both 600 bp and <400 bp bands. For Cre, a positive band at 300 bp was expected, and Cre-negative mice fail to show a band. Embryos that showed floxed null allele at 511 bp for Spred1, a null allele < 400 bp for Spred2, and a Cre-positive band at 300 bp were deemed Spred1/2^CKO^. (^–/–^) Spred1 and Spred2 immunoreactivity was assessed and compared between control (**C**,**E**) and Spred1/2^CKO^ (**G**) transgenic lenses of E14.5 embryos. Spred1 was detected throughout the control lens (**C**), in both the epithelium (le) and primary fiber cells (pf). Spred2 (**E**) was also evident throughout, but with a stronger label in the primary fiber cells (**E**). The Spred1/2^CKO^ failed to show labeling for either Spred1 or Spred2 (**G**). Transgenic lenses were counterstained with Hoechst (**B**,**D**,**F**). Scale bar 100 µm.

**Figure 2 cells-13-00290-f002:**
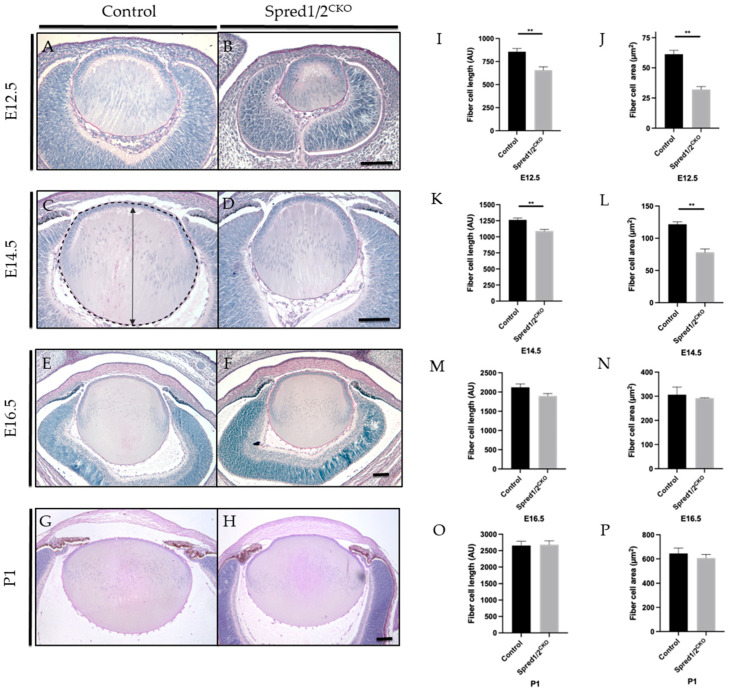
Representative mid-sagittal sections of control and Spred1/2CKO mice lens at E12.5 (**A**,**B**), E14.5 (**C**,**D**), E16.5 (**E**,**F**), and P1 (**G**,**H**) stained with periodic acid-Schiff stain. At E12.5 and E14.5, the Spred1/2CKO lens (**B**,**D**) displayed aberrant lens morphology and small lens phenotype when compared to the control lenses (**A**,**C**). On average, Spred1/2CKO average had significantly shorter fibers at both E12.5 (**I**) 657 ± 36.7 (*n* = 7; *p* = 0.0012) and E14.5 (**K**) 1088 ± 27.71 (*n* = 8, *p* = 0.0013) when compared to control 856 AU ± 34.49 (*n* = 7) and 1264 AU ± 28.01 (*n* = 6), respectively. Spred1/2CKO mice were also found to have significantly smaller total lens area at E12.5 (**J**) 32.06 µm^2^ ± 2.50 (*n* = 7; *p* = 0.0025) and E14.5 (**L**) 78.21 μm^2^ ± 5.207 (*n* = 6, *p* = 0.0022) when compared to the control tissue at 61.31 µm^2^ ± 3.17 (*n* = 5) and 121.6 μm^2^ ± 3.91 *n* = 6, respectively (**J,L**). By E16.5, the Spred1/2CKO lenses (**E**) had recovered in their morphology and fiber cell length (**M**) 1898 AU ± 63.53 (*n* = 6; *p* = 0.0931) and fiber cell area (**N**) 291.8 μm^2^ ± 1.77 (*n* = 6; *p* = 0.6342), compared to control 2122 AU ± 89.28 (*n* = 6) and 306.4 μm^2^ ± 31.84 (*n* = 6), respectively. At P1, Spred1/2CKO lenses (**H**) displayed similar morphology when compared to control lens (**G**), with the exception of an attachment of the lens epithelium to the cornea (**H**). The average lens fiber cell length (**O**) was similar in control 2653 AU ± 133.1 (*n* = 5) and Spred1/2CKO 2681 AU ± 117.1 (*n* = 6, *p* = 0.0931). The average lens fiber area (**P**) was also similar in control 645.2 μm^2^ ± 44.2 (*n* = 5) and Spred1/2CKO 607 μm^2^ ± 30.30 (*n* = 6; *p* = 0.6342), with no significant differences between genotypes (**P**). Panel (**C**) illustrates the measurement of fiber cell length (indicated by double-ended arrow) and lens area (outlined by broken line). Mann–Whitney U test ** *p* < 0.01, error bars represent SEM, scale bars 100 μm.

**Figure 3 cells-13-00290-f003:**
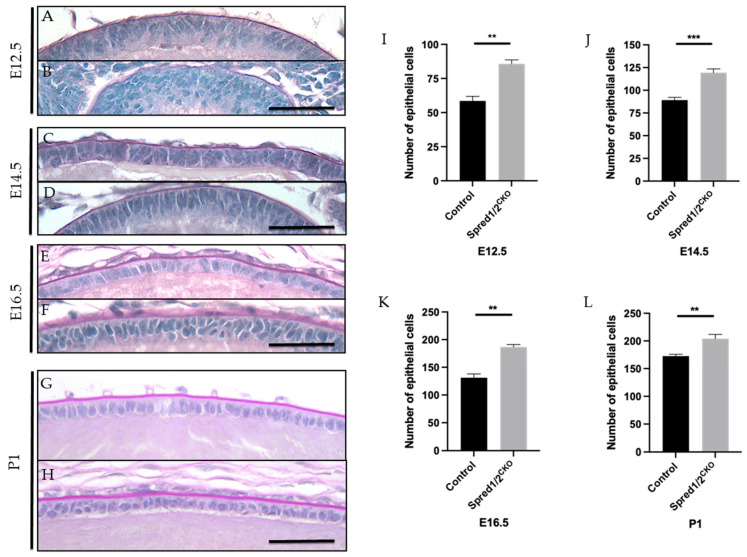
Representative mid-sagittal sections of control and Spred1/2null mice lens at E12.5 (**A**,**B**), E14.5 (**C**,**D**), E16.5 (**E**,**F**), and P1 (**G**,**H**) stained with periodic acid-Schiff stain. The central lens epithelium had a higher cell density and displayed abnormal multilayering of epithelial cells in the Spred1/2^CKO^ lenses across all age groups (**B**,**D**,**F**–**H**) when compared to the control lens (**A**,**C**,**E**,**G**). Quantitative analysis of the number of epithelial cells in the epithelium of mid-sagittal sections was collated. The average number of epithelial cells at E12.5 (**I**) was as follows: control 58.50 ± 3.37 (*n* = 6), Spred1/2^CKO^ 85.67 ± 2.94 (*n* = 6, *p* = 0.0022). At E14.5 (**J**): control 89.17 ± 3.03 (*n* = 6) and Spred1/2^CKO^ 119.3 ± 4.23 (*n* = 10, *p* = 0.0005). At E16.5 (**K**): control 131.2 ± 6.85 (*n* = 6), Spred1/2^CKO^ 186.7 ± 4.43 (*n* = 6, *p* = 0.0022. At P1 (**L**): control 177.6 ± 3.28 (*n* = 5), Spred1/2^CKO^ 204 ± 7.75 (*n* = 5, *p* = 0.0079). There was a significant increase in the number of epithelial cells in the Spred1/2^CKO^ epithelium across all age groups when compared to control lens. Mann–Whitney U test ** *p* < 0.01, *** *p* < 0.001. AU—arbitrary units. Error bars represent SEM, scale bars 50 μm.

**Figure 4 cells-13-00290-f004:**
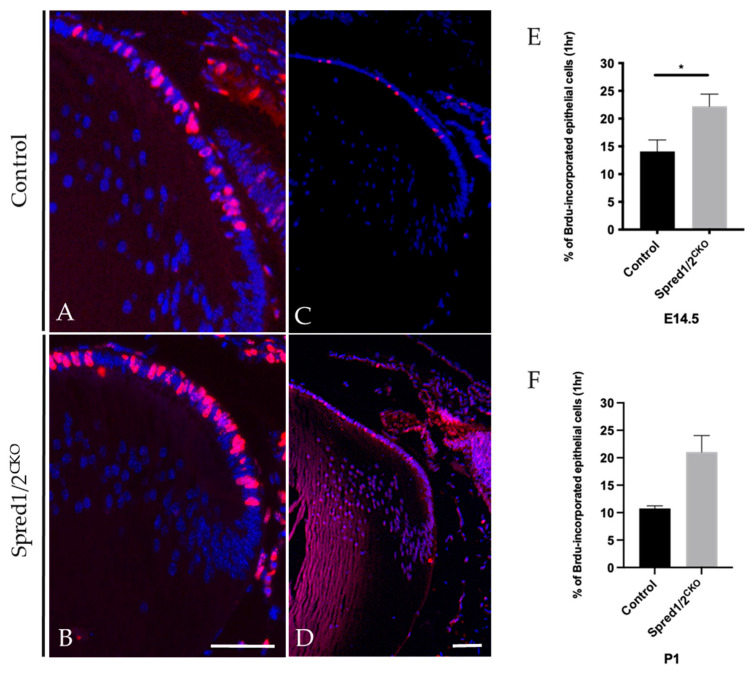
BrdU incorporation of proliferating lens epithelial cells of transgenic and control mice lenses E14.5 (**A**,**B**) and P1 (**C**,**D**) for 1 h. Control and Spred1/2^CKO^ lens tissue at E14.5 was immunolabeled for BrdU incorporated cells (red nuclei) and counterstained with Hoechst. Within 1 h of BrdU injection, there was a significant increase in the number of BrdU incorporated cells in the E14.5 Spred1/2^CKO^ lenses (**B**) when compared to control (**A**). Quantitative analysis revealed a significant increase in the rate of cell proliferation in E14.5 (**E**) Spred1/2^CKO^ lenses at 22.2% ± 2.81(*n* = 9; *p* = 0.021) compared to control at 14.1% ± 2.07 (*n* = 12). An increase in BrdU-positive cells could also be seen postnatally at P1 in Spred1/2^CKO^ (**D**) when compared to control (**C**), where the rate of cell proliferation (**F**) in Spred1/2^CKO^ lenses was at 21% ± 3.0 (*n* = 2), compared to control at 10% ± 0.47 (*n* = 4) (Mann–Whitney U test * *p* < 0.05, error bars represent SEM, scale bar 50 µm).

**Figure 5 cells-13-00290-f005:**
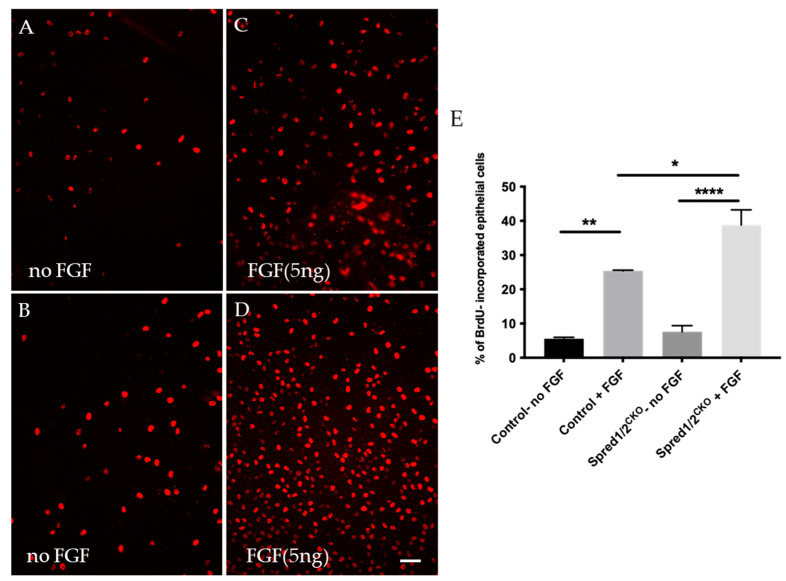
BrdU incorporation of proliferating lens epithelial cells of transgenic and control mice explants at P10. Lens epithelial explants of control (**A**,**C**) and Spred1/2^CKO^ (**B**,**D**) explants were immunolabeled for BrdU (red cells) and were cultured 24 h in vitro. Lens explants were treated either without FGF (controls; (**A**,**B**) or with a low proliferating dose (5 ng/mL) of FGF2 (**C**,**D**). Quantitative analysis of cell proliferation was calculated by dividing the number of BrdU incorporated cells (red) by the total number of cells. Mean epithelial cell proliferation (**E**) of control explants without and with FGF2 was 5.6% ± 0.39 (*n* = 3) and 25.4% ± 0.22 (*n* = 3), respectively. In the Spred1/2^CKO^ mice with and without FGF, this was 7.6% ± 1.8 (*n* = 3) and 38.8% ± 4.4 (*n* = 3), respectively. Cell proliferation was significantly different between the control explants with and without FGF addition (*p* = 0.0017 **), between Spred1/2^CKO^ explants with and without FGF addition (*p* ≤ 0.0001 ****), as well as between the control and Spred1/2^CKO^ explants both treated with FGF (*p* = 0.0170 *). Ordinary one-way ANOVA * *p* < 0.05, ** *p* < 0.01, **** *p* < 0.0001, error bars represent SEM. Scale bar 50 µm.

**Figure 6 cells-13-00290-f006:**
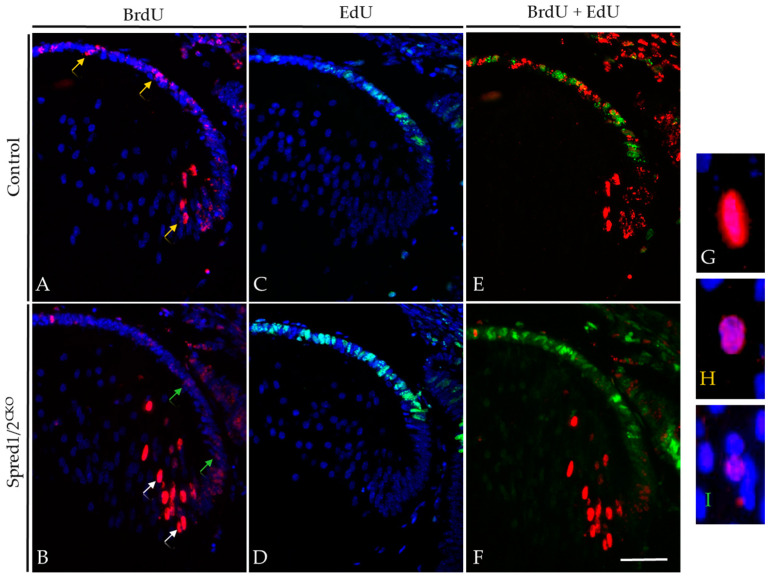
BrdU tracer assay and EdU incorporation in transgenic mice lens at E12.5 + 2. Control (**A**,**C**,**E**) and Spred1/2^CKO^ lenses (**B**,**D**,**F**) at E12.5 + 2 were double-immunolabeled for BrdU (red) and EdU (green) incorporated cells. Differentiating cells were traced over 2 days with BrdU, and proliferating cells were tagged 1 h prior with EdU in both control (**A**,**C**) and Spred1/2^CKO^ lenses (**B**,**D**). When merged, control (**E**) tissue displayed a more even distribution of BrdU incorporated cells, with a moderate level of proliferation, indicated by the EdU incorporated cells. Alternatively, the Spred1/2^CKO^ lens (**F**) had an uneven distribution. The Spred1/2^CKO^ lenses showed BrdU incorporated cells to be primarily in the fiber cell region, with a large population of proliferating cells in the epithelium. White arrows indicate a strong BrdU label, as seen in (**G**), yellow arrows indicate medium BrdU intensity, as seen in (**H**), and green arrows indicate a weak BrdU label, seen in (**I**). Scale represents 50 µm.

**Figure 7 cells-13-00290-f007:**
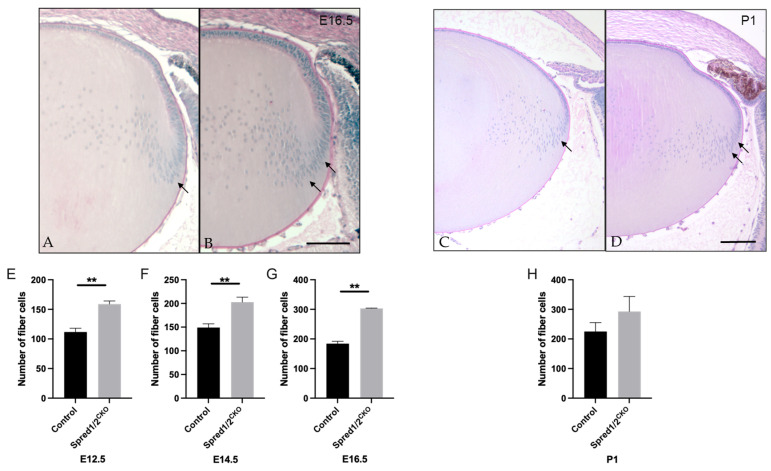
Representative mid-sagittal sections of the fiber cell regions of control and transgenic mice lenses stained with periodic-acid Schiff stain. The fiber cell region had a higher cell density in the Spred1/2^CKO^ (**B**,**D**) mice lens (E16.5 and P1 shown) when compared to control (**A**,**C**), particularly in the bowzone region and newly formed fiber cell region (indicated by arrows). Quantitative analysis of the number of fiber cells of mid-sagittal sections was collated. The average number of fiber cells was as follows: at E12.5 (**E**), control 112 ± 6.49 (*n* = 6) and Spred1/2^CKO^ 158 ± 2.71 (*n* = 6, *p* = 0.0022); at E14.5 (**F**), control 149 ± 7.92 (*n* = 6) and Spred1/2^CKO^ 203 ± 10.42 (*n* = 9; *p* = 0.0028); at E16.5 (**G**), control 184 ± 8.258 (*n* = 6) and Spred1/2^CKO^ 303 ± 1.095 (*n* = 6; *p* = 0.0022). Postnatally (**H**), there was no significant difference in the fiber cell numbers between control (225 ± 30.25, *n* = 9) and Spred1/2^CKO^ lenses (292.7 ± 50.71, *n* = 7). (Mann–Whitney U test ** *p* < 0.01, error bars represent SEM, scale bars 100 µm.

**Figure 8 cells-13-00290-f008:**
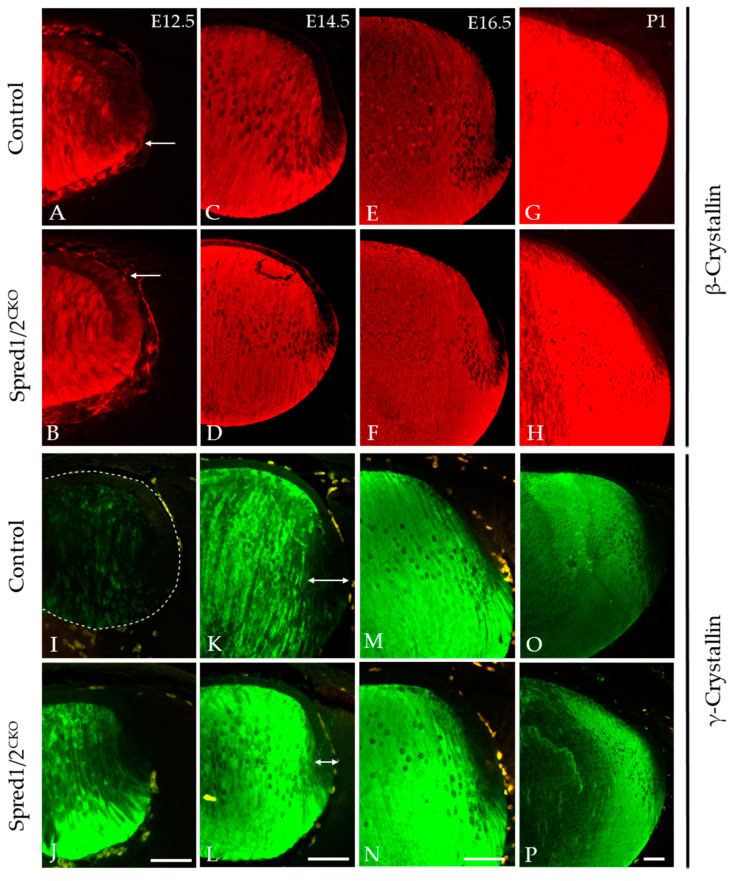
Immunolabeling for β-crystallin and γ-crystallin in control and transgenic mice lens at E12.5, E14.5, E16.5, and P1. β-crystallin localization was investigated in control (**A**,**C**,**E**,**G**) and Spred1/2^CKO^ (**B**,**D**,**F**,**H**) embryonic sections. β-crystallin in the Spred1/2^CKO^ mice was highly expressed and restricted to the fiber cells in the posterior region of the lens at ages E14.5–P1 (**D**,**F**,**H**) and had similar expression when compared to the control (**C**,**E**,**G**). However, at E12.5, in the Spred1/2^CKO^ lens (**B**), β-crystallin levels appeared earlier and relatively stronger in the newly formed fiber cell region (arrows), with elevated basal levels detected in the lens epithelium, when compared to age-matched control (**A**). γ-crystallin localization in control (**I**,**K**,**M**,**O**) and Spred1/2^CKO^ (**J**,**L**,**N**,**P**) embryonic sections. γ-crystallin in the control lens is initially weak at the E12.5 (**I**) age of development, with low expression in the fiber cells (dotted line highlights unlabeled lens). At E14.5 (**K**), expression levels have increased to the whole fiber cell region, and its levels were further increased at E16.5 (**M**) and at P1 (**O**). Alternatively, Spred1/2^CKO^ mice showed high expression earlier at E12.5 (**J**), markedly higher than that of the control (arrows). This was also the case at E14.5 (**L**), E16.5 (**N**), and P1 (**P**). Scale bars, 50 μm.

**Figure 9 cells-13-00290-f009:**
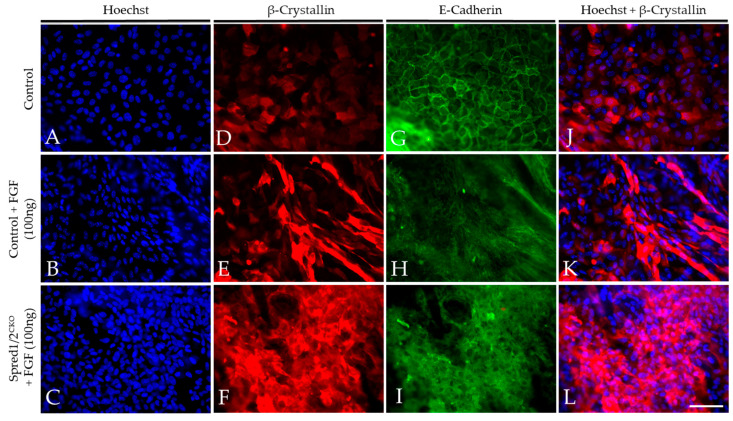
Representative micrographs of β-crystallin and E-cadherin in the explants of WT and Spred1/2^CKO^ FGF2 (100 ng/mL) cultured lenses at P10. Explants were double labeled for both β-crystallin (**D**–**F**,**J**–**L**) and E-cadherin (**G**–**I**) and were counter stained with Hoechst (**A**–**C**,**J**–**L**) after a 10-day treatment with a high dose of FGF2 (100 ng/mL). FGF2-treated lens explants showed strong β-crystallin expression in both WT (**E**) and Spred1/2^CKO^ (**F**) whole mounts when compared to basal levels seen in the no-treatment control (**D**). β-crystallin expression appeared to be stronger in Spred1/2^CKO^ (**F**) when compared to WT (**E**). E-cadherin expression was seen to be lost in the FGF2 treatment groups in both WT (**H**) and Spred1/2^CKO^ (**I**) explants. The non-treated control (**G**) retained E-cadherin expression in the cell membrane. Scale bar, 50 µm.

**Figure 10 cells-13-00290-f010:**
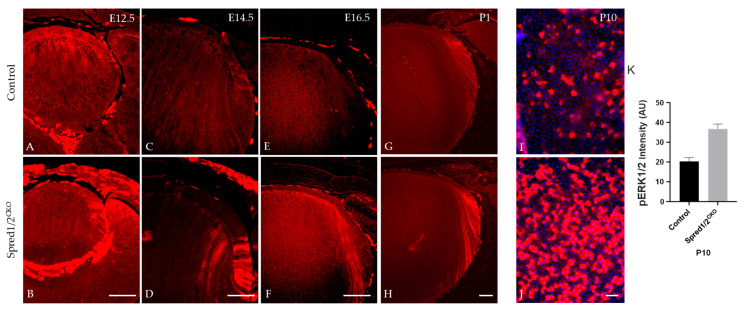
Immunolabeling for pERK in transgenic mice lens at E12.5, E14.5, E16.5, and P1. pERK immunoreactivity was investigated in control (**A**,**C**,**E**,**G**,**I**) and Spred1/2^CKO^ mice lenses (**B**,**D**,**F**,**H**,**J**). Spred1/2^CKO^ showed significantly increased levels of pERK, particularly in the germinative and transitional regions at ages E12.5 (**B**) and E14.5 (**D**), when compared to control ((**A**,**C**) respectively). At E16.5 and P1, Spred1/2^CKO^ tissue (**F**,**H**) had increased levels, particularly in the transitional and newly forming primary fiber cell regions, when compared to control lens (**E**,**G**). P10 epithelial explants shows a singificant increase in pERK expression in Spred1/2^CKO^ explants (**J**) when compared to control (**I**). (**K**) Quantitative analysis confirms this observation, with Spred1/2^CKO^ lenses exhibiting higher pERK1/2 levels (36.71 ± 2.42 AU, *n* = 5) compared to controls (20.30 ± 1.92 AU, *n* = 5; *p* = 0.0079), Mann–Whitney U test, AU—arbitrary units; error bars represent SEM. Scale bars, 50 μm.

**Figure 11 cells-13-00290-f011:**
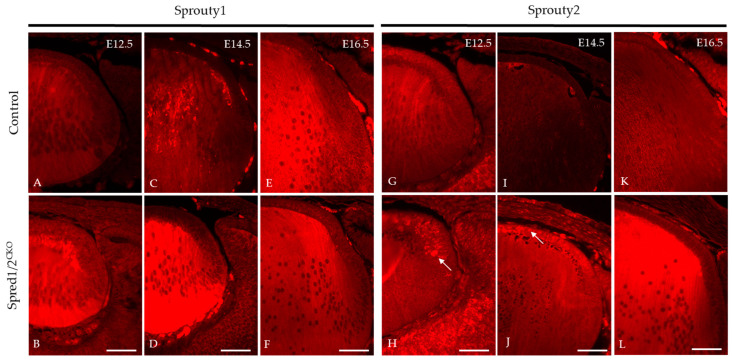
Immunolabeling of Sprouty1 and Sprouty2 expression in transgenic mice lens at E12.5, E14.5, and E16.5. Sprouty1 levels were investigated in control (**A**,**C**,**E**) and Spred1/2^CKO^ (**B**,**D**,**F**) sections. Sprouty1 immunoreactivity was seen to be expressed throughout the whole lens. Spred1/2^CKO^ showed significantly increased levels of Sprouty1 at ages E12.5 (**B**) and E14.5 (**D**), particularly in the fiber cell regions, when compared to control (**A**,**C**) respectively. At E16.5, levels of Sprouty1 in Spred1/2^CKO^ (**F**) lens appeared to be consistent with the control (**E**) lens. Sprouty2 immunoreactivity was seen to be expressed throughout the whole lens. Spred1/2^CKO^ showed markedly increased levels of Sprouty2 at ages E12.5 (**H**) and E14.5 (**J**), particularly in the epithelium in the form of clusters (indicated by arrows), when compared to control (**G**,**I**) respectively. At E16.5, levels of Sprouty2 in Spred1/2^CKO^ (**L**) appeared to be markedly increased in the fiber cell region of the lens when compared to control (**K**). Scale bars, 50 μm.

## Data Availability

The raw data supporting the conclusions of this article will be made available by the authors on request.

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
