# Peer review of "Conditional Ablation of Spred1 and Spred2 in the Eye Lens Negatively Impacts Its Development and Growth"

_cells, 2024, doi:10.3390/cells13040290_

Round 1

Reviewer 1 Report

Comments and Suggestions for Authors

This paper seeks to enhance our knowledge of the role(s) of Spreds in regulating prenatal lens epithelial cell proliferation and fiber differentiation. The authors generate and characterize conditional knockout mice lacking both Spred1 and Spred2. The cKO lenses display elevated ERK1/2 phosphorylation, hyperproliferation of lens epithelia, and an increase in the rate of fiber differentiation. Paradoxically, this results in transient microphakia and microphthalmia, which later recovers, possibly due to compensatory upregulation of Sprouty expression. Overall, this is an interesting study with well-designed experiments. I have a few comments and suggestions for improvement.

1. The authors report that increased epithelial cell proliferation and fiber cell differentiation result in a smaller lens. They also state that they saw no evidence of cell death in the double cKO lenses. This is quite a paradox, which was not reviewed in any depth in the discussion. If there are more cells being produced and differentiated, and yet the lens is smaller. There must be a logical explanation.

 2. Page 5, line 214. “we” should be “were”, i.e. “embryos were”.

 3. In section 3.3, are the average number of epithelial cells being counted in the field of view, or across the entire epithelium? Also, in the legend to Figure 3 the First sentence is garbled.

 4. In section 3.9, was the decrease in pERK1/2 phosphorylation verified by western blot?

Author Response

  1. The authors report that increased epithelial cell proliferation and fiber cell differentiation result in a smaller lens. They also state that they saw no evidence of cell death in the double cKO lenses. This is quite a paradox, which was not reviewed in any depth in the discussion. If there are more cells being produced and differentiated, and yet the lens is smaller. There must be a logical explanation.

We have added further Discussion on this paradox. See lines 642-655.

  1. Page 5, line 214. “we” should be “were”, i.e. “embryos were”.

Thank you for identifying this. We have corrected it.

  1. In section 3.3, are the average number of epithelial cells being counted in the field of view, or across the entire epithelium?

In Section 3.3, the average number of epithelial cells were counted across the entire epithelium, and not simply the field of view, as stated in the Methods.

Also, in the legend to Figure 3 the First sentence is garbled.

Thank you for identifying this. We have corrected it.

  1. In section 3.9, was the decrease in pERK1/2 phosphorylation verified by western blot?

We unfortunately were not able to validate the decrease in pERK1/2 phosphorylation using western blots as we could not cleanly isolate lens tissues from these early developmental stages, without the risk of contaminating extra-lenticular tissues. With that said, we were meticulous in ensuring that all experimental procedures were constant across all tissues from immunolabeling to image capture.

Reviewer 2 Report

Comments and Suggestions for Authors

In this manuscript, the authors report the role of Spred1 and Spred2 as suppresses of ERKs signaling to control lens epithelial cell proliferation and differentiation during embryonic lens development. Using lens specific Spreds conditional knockout (cKO) mice, the authors reveal solid results of reduced embryonic lens size at E12.5-14.5 dpc but the lens size eventually recovered after E16.5 dpc. Spreds cKO lenses display higher rate of lens epithelial cell proliferation with appearance of multiple EP layers and increased rate of fiber differentiation with earlier expression of beta- and gamma-crystallins but reduced fiber elongation. The authors further provided a loss of Spreds led to increased pERKs. Therefor, the mechanistic conclusions are sounds that Spreds function as antagonism of ERKs to regulate lens cell proliferation and differentiation.  The data quality is excellent. There are two minor concerns about data clarification, interpretation or rationale are listed below.  

Given the fact that Spreds cKO lens recovers normal lens size, does the total number of the lens epithelial (EP)cells (with multiple layers) remain the same as when compared to age-matched wild-type control? If the total EP cell number is unchanged in Spreds cKO lenses, then the role of Spreds seems to regulate the epithelial cell polarity/morphogenesis between monolayer versus multiple cell layers.

The normal size of Spreds cKO lenses show increased rate of fiber cell differentiation with higher fiber cell density (according to the fiber cell nuclei). It’s likely these differentiating fibers are smaller (or shorter) in size due to elongation defects? Perhaps, the authors should show higher resolution images of cKO cortical fibers for clearly elucidating this unique lens growth phenomenon. A loss of Spreds promotes cell differentiation but not elongation. There are not results to address fiber cell maturation in cKO lenses. The authors need to revise the discussion section 4.3. “Spreds negatively regulate lens fiber cell differentiation (elongation and maturation)" 

Author Response

Given the fact that Spreds cKO lens recovers normal lens size, does the total number of the lens epithelial (EP) cells (with multiple layers) remain the same as when compared to age-matched wild-type control? If the total EP cell number is unchanged in Spreds cKO lenses, then the role of Spreds seems to regulate the epithelial cell polarity/morphogenesis between monolayer versus multiple cell layers.

Our data shows that the total number of (multilayered) lens epithelial cells of our mutants is significantly higher than those calculated for control lens for all ages examined (see Figure 3), even after recovery. While we do not discount that Spreds may regulate cell polarity, we do not have any direct support for this but have now alluded to it in the Discussion.

The normal size of Spreds cKO lenses show increased rate of fiber cell differentiation with higher fiber cell density (according to the fiber cell nuclei). It’s likely these differentiating fibers are smaller (or shorter) in size due to elongation defects? Perhaps, the authors should show higher resolution images of cKO cortical fibers for clearly elucidating this unique lens growth phenomenon. A loss of Spreds promotes cell differentiation but not elongation. There are not results to address fiber cell maturation in cKO lenses. The authors need to revise the discussion section 4.3. “Spreds negatively regulate lens fiber cell differentiation (elongation and maturation)" 

While we do not believe that there are lens fiber cell elongation defects, given we see a subsequent recovery of fiber length, we have proposed that the rate of epithelial cell differentiation to secondary fibers is increased; however, the elongation is retarded. Despite this retardation of cell elongation, that we have reported is linked to ERK1/2-signaling (target of Spreds), we do not see a corresponding delay in fiber-specific crystallin accumulation (that we have also previously reported to be independent of ERK1/2-signaling). With that said, we know that for ‘normal’ fiber differentaion, b-crystallin is expressed ahead of g-crystallin, the latter being just one feature of fiber cell maturation. As we observe an early and advanced accumulation of g-crystallin in our mutants, we believe that the impaired fiber differentiation impacts not only the elongation but also cell maturation. We appreciate that additional maturation markers would help resolve this and we respectfully would like to leave the Discussion as originally presented.

Reviewer 3 Report

Comments and Suggestions for Authors

1. Fig 5, Fig 8, Fig 9, Fig 10, Fig 11 lack scale bars.

2. For all images, please use dotted lines to indicate the basement membrane

3. Fig 1A-H, you can combine all of them into 1A, indicating what each column represents for certain genotypes.

4. Fig. 2H is duplicated by image and graph

5. Fig 2H-O, a diagram is needed for measuring the length and the area. 

6. Fig 3B, is this the one you found representative? it does not contain a similar region as in 3A

7. Fig. I-L, I think what you should do is the number of epithelial cells per um (or mm) since you try to indicate the cells are more stratified. 

8. Fig 4 and 5 can be combined

9. Fig 6G-I, you should apply a quantification using fluorescent intensity to support your statement in line 400-409.

10. Fig 7A and B, higher magnification of the target region is required

11. Fig 7C and D, use arrows to point out the target region you mentioned in line 437-441

12. Fig 8A and B, you need fluorescent intensity measurement towards ROIs to address your description in line 459-461.

13, Fig 8I and J, same as comment 12

14. Fig 9 D-F, you need to draw the outline of the cells and show merged with nuclei staining to support your description in line 486-488

15. Fig 9 G-I, you also need to measure fluorescent intensity to correspond with what you mentioned in line 489-491

16. Fig 10 A and B, C and D, same as comment 12 that correspond with your description in line 508-512.  

17. Fig 10 I and J, the result should be quantified by fluorescent intensity of pERK divided by nuclei staining fluorescent intensity. 

18. Fig 11, same as comment 12, you need quantification of fluorescent intensity towards certain ROIs. 

Comments on the Quality of English Language

The overall writing of this manuscript is acceptable but can be further improved.

Author Response

  1. Fig 5, Fig 8, Fig 9, Fig 10, Fig 11 lack scale bars.

We have rectified this and now added scale bars to these Figures.

  1. For all images, please use dotted lines to indicate the basement membrane.

We have added dotted lines to indicate the lens capsule (basement membrane) only in some Figures when it did not obscure the data trying to be demonstrated.

  1. Fig 1A-H, you can combine all of them into 1A, indicating what each column represents for certain genotypes.

We have now altered the labeling for Figure 1 as suggested, to make it simpler to follow.

  1. 2H is duplicated by image and graph

Thank you for identifying this. We have now corrected the labeling for Figure 1 as suggested.

  1. Fig 2H-O, a diagram is needed for measuring the length and the area. 

We have incorporated lines in Figure 2 to support the methodology on where length of fiber cells and area of lens was calculated.

  1. Fig 3B, is this the one you found representative? it does not contain a similar region as in 3A.

Figure 3 shows representative regions of lens epithelia. They are all taken from similar regions of control and mutant lens, as described in methods, using comparable mid-sagittal sections.

  1. I-L, I think what you should do is the number of epithelial cells per um (or mm) since you try to indicate the cells are more stratified. 

Our counts identified total number of epithelial cells across the entire epithelia as described in the methods. While expressing the number of cells relative to distance, this would indeed further highlight the multilayering. As this cell multilayering is very evident in all our histology, compared to the control epithelia that remains as a monolayer, we believe this is sufficient. Had we been comparing different levels of multilayering, the number of cells as a measure of distance may have been more useful as the referee suggests.

  1. Fig 4 and 5 can be combined.

Thank you for this suggestion. When we tried to do this, the individual plates of the combined Figure were simply too small, and the data did not present as well. We have respectfully left it as two Figures.  

  1. Fig 6G-I, you should apply a quantification using fluorescent intensity to support your statement in line 400-409.

Figure 6G-I is similar to a heat map for demonstrating the level of labeling (incorporation of BrdU); stonger labeling of nuclei is presented as red, and weakest incorporation of BrdU/labeling, as blue (Hoechst label). The differences are simply a guide and we have not quantified this.

  1. Fig 7A and B, higher magnification of the target region is required

We do not have higher resolution images of these regions to present. We have enlarged the image to allow a better view of the target region. For those using digital media to review this manuscript they can readily enlarge it as needed.

  1. Fig 7C and D, use arrows to point out the target region you mentioned in line 437-441

Thank you for this suggestion. We have now added arrows to better highlight these regions.

  1. Fig 8A and B, you need fluorescent intensity measurement towards ROIs to address your description in line 459-461.

We are not able to confidently quantify this.

13, Fig 8I and J, same as comment 12.

We are not able to confidently quantify this.

  1. Fig 9 D-F, you need to draw the outline of the cells and show merged with nuclei staining to support your description in line 486-488

We have now merged some of these images for Figure 9, to better highlight both fiber cells and lens epithelial cells with respect to cell nuclei.

  1. Fig 9 G-I, you also need to measure fluorescent intensity to correspond with what you mentioned in line 489-491

We are not able to confidently quantify this.

  1. Fig 10 A and B, C and D, same as comment 12 that correspond with your description in line 508-512.

We are not able to confidently quantify this.

  1. Fig 10 I and J, the result should be quantified by fluorescent intensity of pERK divided by nuclei staining fluorescent intensity.

For this Figure, we were able to quantify the fluorescent intensity of pERK shown in lens epithelial explants. See Fig 10K.

  1. Fig 11, same as comment 12, you need quantification of fluorescent intensity towards certain ROIs.

We are not able to confidently quantify this.

Round 2

Reviewer 3 Report

Comments and Suggestions for Authors

The overall quality of the manuscript has been improved after revised.

There are two concerns that remain in the current version: 

1. If there is increased proliferation and differentiation w/o cell death found, how would authors explain the formation of the smaller lens?

2. My suggestions for measuring fluorescent intensity have not been addressed in the last revision. FIJI is a useful tool that can measure the fluorescent intensity in ROI. 

Comments on the Quality of English Language

It's acceptable for the journal publication.

Author Response

  1. If there is increased proliferation and differentiation w/o cell death found, how would authors explain the formation of the smaller lens?

We thank the reviewer for the new questioning of this. We have already addressed this in the revised manuscript in response to another reviewers’ comments. Please see the Discussion.

  1. My suggestions for measuring fluorescent intensity have not been addressed in the last revision. FIJI is a useful tool that can measure the fluorescent intensity in ROI. 

We too would ideally have liked to measure and quantified the fluorescent intensity in ROI as requested. It is not that we ignored the reviewers request or that we did not want to address this. The fact is that to do this effectively in the tissue sections, and with the confidence that we expect, it is very difficult. As the reviewer can imagine, these images were collected over a considerable time period from many different animals, and we primarily captured representative images from countless lenses. To quantify the fluorescence at this late stage, we only have set images to work with, and no longer have the primary tissue where we would have liked to have measured the intensity of the whole lenses, and not just ROI. While we propose differences to levels of specific proteins, for many it is the patterns that differ. We do not state anywhere that these changes in levels are significant to any degree. I appreciate it would be good to gain a measure here to show some significance, but we are limited with the materials we currently have at hand. Thank you for your understanding.